# Actinide-lanthanide single electron metal-metal bond formed in mixed-valence di-metallofullerenes

Yingjing Yan[1,7], Laura Abella[2,7], Rong Sun[3,7], Yu-Hui Fang ®[3], Yannick Roselló ®[2], Yi Shen[1], Meihe Jin[1], Antonio Rodríguez-Fortea ®[2], Coen de Graaf ®[2,4], Qingyu Meng[1], Yang-Rong Yao ®[1,5] ✉, Luis Echegoyen[6], Bing-Wu Wang ®[3] ✉, Song Gao ®[3], Josep M. Poblet ®[2] ✉ & Ning Chen ®[1] ✉

Understanding metal-metal bonding involving f-block elements has been a challenging goal in chemistry. Here we report a series of mixed-valence di-metallofullerenes, ThDy@C$_{2n}$ ($2n$ = 72, 76, 78, and 80) and ThY@C$_{2n}$ ($2n$ = 72 and 78), which feature single electron actinide-lanthanide metal-metal bonds, characterized by structural, spectroscopic and computational methods. Crystallographic characterization unambiguously confirmed that Th and Y or Dy are encapsulated inside variably sized fullerene carbon cages. The ESR study of ThY@$D_{3h}$(5)-C$_{78}$ shows a doublet as expected for an unpaired electron interacting with Y, and a SQUID magnetometric study of ThDy@$D_{3h}$(5)-C$_{78}$ reveals a high-spin ground state for the whole molecule. Theoretical studies further confirm the presence of a single-electron bonding interaction between Y or Dy and Th, due to a significant overlap between hybrid spd orbitals of the two metals.

Metal-metal bonding is a classic research topic in chemical bonding studies and has been applied as a tool for developing molecular magnets as well as for addressing challenges in biology, energy, and catalysis[1,2]. Thus far, most of the metal-metal bonding studies have focused on the interactions involving d orbitals. By contrast, direct bonds between f-block metals are extremely difficult to prepare by conventional synthetic methods due to the limited extension of 4f or 5f orbitals, and remained elusive until the very recent isolation of a dilanthanide complex featuring lanthanide metal-metal bonds[3]. In this study, Long, Harvey, and Chilton et al. significantly demonstrated that single electron lanthanide metal-metal bonds, which give rise to an enormous coercive magnetic field at liquid nitrogen temperature, can be obtained in mixed-valence dilanthanide complexes (Cp$^{iPr5}$)$_2$Ln$_2$I$_3$ (Ln

= Gd, Tb, or Dy) via salt metathesis reaction[4]. On the other hand, Liddle et al. also reported the synthesis of a tri-thorium cluster with a delocalized 3-center-2-electron Th-Th bond very recently by using K$_2$[C$_4$(SiMe$_3$)$_4$] as the reduced reagent to generate thorium(III)-containing complexes which contains low-valence thorium ions in close proximity[5].

Endohedral doping of fullerenes with a variety of metal atoms or metallic clusters to form endohedral metallofullerenes (EMFs) provides many possibilities for the investigation of metal-metal interactions[6–8]. In particular, di-metallofullerenes (di-EMFs), with only two metal atoms trapped inside the carbon cages, i.e. M$_2$@C$_{2n}$, provide a unique platform to study these bonding interactions. In these di-EMFs, the two metal atoms generally adopt relatively long

[1]College of Chemistry, Chemical Engineering and Materials Science, and State Key Laboratory of Radiation Medicine and Protection, Soochow University, Suzhou, Jiangsu 215123, P. R. China. [2]Departament de Química Física i Inorgànica, Universitat Rovira i Virgili, Marcel·lí Domingo 1, 43007 Tarragona, Spain. [3]Beijing National Laboratory for Molecular Sciences, State Key Laboratory of Rare Earth Material Chemistry and Application, College of Chemistry and Molecular Engineering, Peking University, Beijing 100871, P. R. China. [4]ICREA, Pg. Lluís Companys 23, 08010 Barcelona, Spain. [5]Department of Materials Science and Engineering, University of Science and Technology of China, Hefei 230026, P. R. China. [6]Department of Chemistry, University of Texas at El Paso, 500 W University Avenue, El Paso, TX 79968, USA. [7]These authors contributed equally: Yingjing Yan, Laura Abella, Rong Sun. ✉e-mail: yryao@ustc.edu.cn; wangbw@pku.edu.cn; josepmaria.poblet@urv.cat; chenning@suda.edu.cn

metal-metal distances, dictated by the metal-cage interactions as well as by the repulsion between the partially ionized atoms due to electron density transfer to the carbon cages[9–13]. However, despite the long distances and the repulsion forces, the metal dimers cannot be dissociated inside the nano-scale fullerene cage, thus direct lanthanide metal-metal bonds, which are hardly accessible by conventional synthetic methods so far, can form inside the fullerene cages[14]. Recently, much progress has been made in the study of lanthanide metal-metal bonds inside fullerene cages[15–19]. In addition to the $\sigma^2$ lanthanide metal-metal bond found for $Lu_2$, $Er_2$, and $Sc_2$ inside the $C_{3v}(8)$-$C_{82}$ fullerene cage[15,20], single electron lanthanide metal-metal bonds have attracted much attention because of their potential in the preparation of novel molecular magnets. A particular array of lanthanide metal dimers, i.e., $Y_2$, $Dy_2$, $Gd_2$, $Tb_2$, etc. were found to form single-electron bonds inside fullerene cages[19,21]. These di-EMFs were synthesized by arc-discharge method similar to those of other EMFs, but their carbon cages are either doped with N or attached with functional groups to obtain the stable compound in solution[22,23]. These results offered an evidence of single electron lanthanide metal-metal bonds and also showed that the outstanding single molecular magnetism of the corresponding di-EMFs arises from this unique bonding interaction[18,19,24,25].

Metal-metal bonding between f-block elements, actinides, have also been proposed and studied by theoreticians[26–30]. Similar to the lanthanides, the synthesis and characterization of a molecular compound containing an actinide-actinide bond is still a great challenge[5,31]. Our recent studies show that two U atoms, as predicted by computational studies[32], can be encapsulated inside an $I_h(7)$-$C_{80}$ cage to form a stable di-metallic actinide EMF $U_2@I_h(7)$-$C_{80}$[33]. Moreover, the two low oxidation states of Th(III) were found to form a strong $\sigma^2$ single actinide-actinide metal-metal bond in $Th_2@I_h(7)$-$C_{80}$[34]. These studies provide the experimental proof of actinide-actinide metal-metal bonds in a molecular compound and again demonstrates the advantage of using fullerenes as template structures to study these elusive metal-metal bonds involving f elements.

We wondered if this paradigm could be further extended to heteronuclear di-metallofullerene compounds with direct metal-metal bonding between a lanthanide and an actinide. The possibility of obtaining and characterizing lanthanide-actinide metal-metal bonds is not only interesting for endohedral fullerene studies, but more importantly, they may offer a potentially powerful platform for the fundamental understanding of f-block elemental chemistry.

Herein, we report the formation of unprecedented actinide-lanthanide single electron metal-metal bonds inside fullerene cages, giving rise to a series of mixed di-metallofullerenes, $ThDy@C_{2n}$ ($2n$ = 72, 76, 78, and 80) and $ThY@C_{2n}$ ($2n$ = 72 and 78). These mixed actinide-lanthanide di-EMFs were synthesized and characterized by single-crystal X-ray diffraction and multiple spectroscopic methods. In particular, SQUID magnetometry, ESR spectra, and computational analyses were performed to probe the actinide-lanthanide single-electron metal-metal bonding.

## Results
### Synthesis and isolation of ThX@C$_{2n}$ (X = Dy and Y; $2n$ = 72, 76, 78, and 80)
$ThDy@C_{2n}$ ($2n$ = 72, 76, 78, and 80) and $ThY@C_{2n}$ ($2n$ = 72 and 78) were synthesized by a modified arc-discharge method. Graphite, mixed with $ThO_2$ and $Dy_2O_3/Y_2O_3$ (molar ratio of Th:Dy:Y:C = 1:1:24) were evaporated in a 200 Torr He atmosphere using a current of 90 A. The resulting soot was extracted with $CS_2$ for 12 h. A multistage HPLC procedure was employed to isolate and purify the $ThDy@C_{2n}$ and $ThY@C_{2n}$ samples (Supplementary Figs. 2–8). The positive-ion mode MALDI-TOF mass spectra of purified $ThDy@C_{2n}$ ($2n$ = 72, 76, 78, and 80) and $ThY@C_{2n}$ ($2n$ = 72 and 78) (Supplementary Fig. 1) show single peaks at m/z = 1260.091, 1308.092, 1332.103, 1356.097, 1184.960, and 1256.961,

respectively, and the experimental isotopic distributions agree well with the calculated ones. The purities of $ThDy@C_{2n}$ and $ThY@C_{2n}$ were confirmed by the observation of HPLC single peaks (Supplementary Fig. 1).

### Molecular structures of ThX@C$_{2n}$·[Ni$^{II}$(OEP)] (X = Dy and Y; $2n$ = 72, 76, 78, and 80)
The molecular structures of the obtained EMFs were determined by single-crystal X-ray diffraction studies. Black crystals with suitable sizes were obtained by slow diffusion from a benzene solution of $Ni^{II}$(OEP) (OEP = 2, 3, 7, 8, 12, 13, 17, 18-octaethylporphyrin anion) into a $CS_2$ solution of the EMFs. As shown in Figs. 1 and 2, the six EMFs were all co-crystallized with one $Ni^{II}$(OEP) molecule, with substantial $\pi$–$\pi$ interactions between them, and unambiguously refined as $ThDy@D_2(10611)$-$C_{72}$, $ThDy@C_s(17490)$-$C_{76}$, $ThDy@D_{3h}(5)$-$C_{78}$, $ThDy@I_h(7)$-$C_{80}$, $ThY@D_2(10611)$-$C_{72}$, and $ThY@D_{3h}(5)$-$C_{78}$. Among them, $ThDy@D_2(10611)$-$C_{72}$ and $ThY@D_2(10611)$-$C_{72}$ share the same non-isolated pentagon rule (non-IPR) fullerene cage, $D_2(10611)$-$C_{72}$[9]. The cages of $C_s(17490)$-$C_{76}$ and $I_h(7)$-$C_{80}$ are fully ordered, and the other four cages are disordered over two orientations (cage A and cage B), as listed in Supplementary Table 1. The metal ions inside all the fullerene cages also show some disorder. Interestingly, due to the strong metal-pentalene interactions, the metal ions in the three non-IPR cages i.e. $D_2(10611)$-$C_{72}$ and $C_s(17490)$-$C_{76}$, show less disorder than in the three IPR cages. Details of the metallic occupancies are summarized in Supplementary Table 2. Only the major cage orientations and metal sites are discussed below.

Figures 1 and 2 illustrate the enlarged regions showing the interactions between the encapsulated Th/Ln (Ln = Dy and Y) units and their corresponding cage moieties and for the dimers themselves. The Th/Ln ions are located over the [5, 5] bonds in non-IPR $C_{72}$ and $C_{76}$ cages and two parallel symmetrical hexagons in the IPR $C_{78}$ and $C_{80}$ cages. The lanthanide ions have slightly shorter metal-cage distances than the Th ions (Supplementary Tables 3 and 4). Th-cage distances are very close for all six fullerene cages, indicating that there is little influence of the cage size and shape on the metal-cage interactions. Further structural analyses show that the Th/Dy and Th/Y units have similar metal-metal distances when encapsulated in the same cage, 4.166(3)/4.183(1) Å in $D_2(10611)$-$C_{72}$ and 4.135(9)/4.144(8) Å in $D_{3h}(5)$-$C_{78}$. The results suggest that the lanthanide ions have a minor effect on the Th-Ln interactions, which may result from the similar lanthanide ionic radii, $Y^{3+}$(0.90 Å) vs. $Dy^{3+}$(0.91 Å)[35]. However, the Th/Dy unit has a metal-metal separation of 3.932(7) Å in $I_h(7)$-$C_{80}$, which is notably shorter than those in $D_2(10611)$-$C_{72}$ (4.166(3) Å), $C_s(17490)$-$C_{76}$ (4.193(2) Å), and $D_{3h}(5)$-$C_{78}$ (4.135(9) Å). This correlates with the fact that the longest axis of 7.84 Å inside the $I_h(7)$-$C_{80}$ is much shorter than those of the $D_2(10611)$-$C_{72}$ (8.87 Å), $C_s(17490)$-$C_{76}$ (8.58 Å), and $D_{3h}(5)$-$C_{78}$ (8.03 Å). This indicates that, while metal-cage distances are identical for the different cages, the Th-Ln interactions are largely cage-dependent.

It is noteworthy that the Th/Dy distance in $I_h(7)$-$C_{80}$ of 3.932(7) Å is comparable to those of the $Dy_2$ dimer in $Dy_2@C_{80}(CH_2Ph)$ (3.898(3) Å)[18] and $Dy_2@C_{79}N$ (3.89 Å)[36] with a two-center single-electron (2c-1e) bond, but longer than that of the $Th_2$ dimer that contains a covalent Th-Th bond in the same cage (3.816 Å)[34]. It might indicate that the Th/Dy interaction is likely comparable to those for $Dy_2@C_{80}(CH_2Ph)$ (3.898(3) Å)[18] and $Dy_2@C_{79}N$ (3.89 Å)[36] with a 2c-1e single-electron bond.

### Electronic structures of ThX@C$_{2n}$ (X = Dy and Y; $2n$ = 72, 76, 78, and 80) and electrochemical properties
DFT calculations show a spin-septet configuration for $ThDy@D_2(10611)$-$C_{72}$, $ThDy@C_s(17490)$-$C_{76}$, $ThDy@D_{3h}(5)$-$C_{78}$ and $ThDy@I_h(7)$-$C_{80}$ (Supplementary Fig. 10). The optimized Th-Dy distances, 4.187 Å for $ThDy@D_2(10611)$-$C_{72}$, 4.209 Å for $ThDy@C_s(17490)$-$C_{76}$, 4.151 Å for

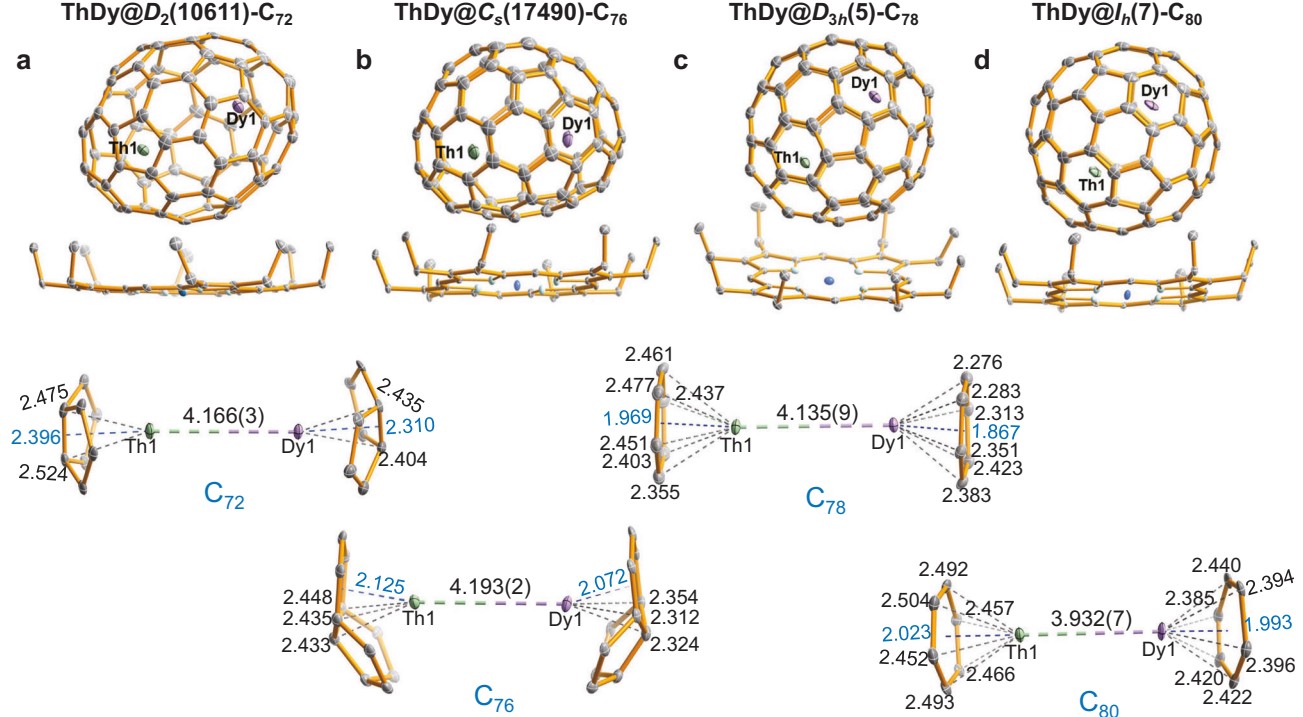

**Fig. 1 | Single-crystal X-ray diffraction studies for ThDy@C$_{2n}$ (2n = 72, 76, and 80).** ORTEP drawings of **a** ThDy@$D_2$(10611)-C$_{72}$, **b** ThDy@$C_s$(17490)-C$_{76}$, **c** ThDy@$D_{3h}$(5)-C$_{78}$, and **d** ThDy@$I_h$(7)-C$_{80}$ with the co-crystallized Ni$^{II}$(OEP) molecule (upper) and the relative positions of the major Th1/Dy1 dimer to a partial region of the fullerene cage (below). Only the major cage orientations and metal sites, Th1 (green) and Dy1 (purple) are shown. Solvent molecules and hydrogen atoms are omitted for clarity. Metal-cage, metal-centroid, and metal-metal distances (in Å) are indicated.

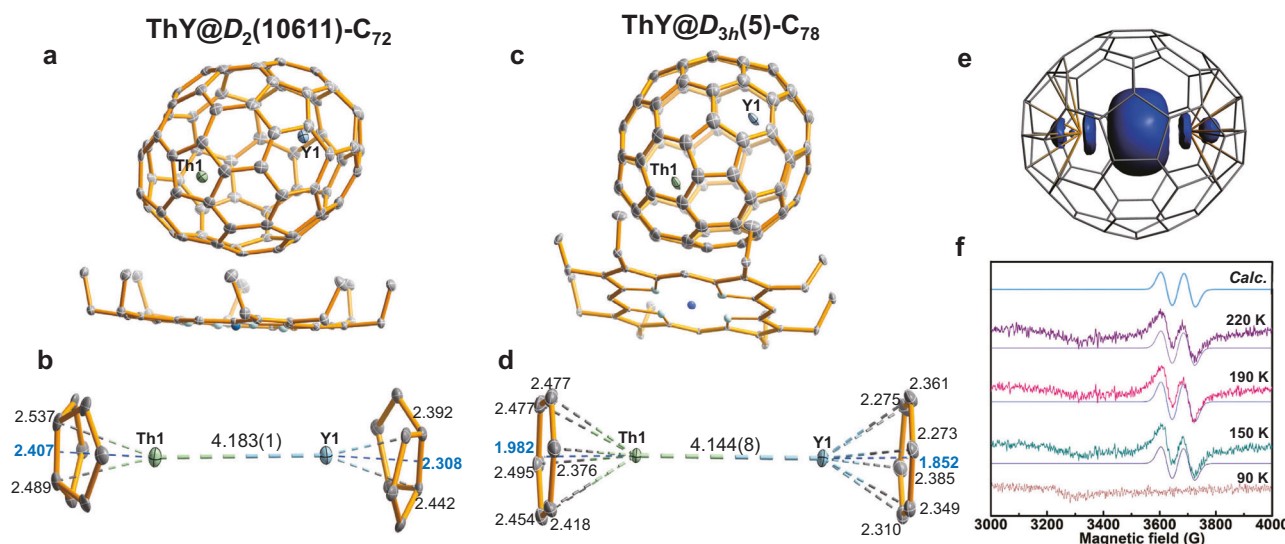

**Fig. 2 | Molecular studies for ThY@C$_{2n}$ (2n = 72 and 78). a** and **c** ORTEP drawings of ThY@$D_2$(10611)-C$_{72}$ and ThY@$D_{3h}$(5)-C$_{78}$ with the co-crystallized Ni$^{II}$(OEP) molecule. Only the major cage orientations and metal sites, Th1 (green) and Y1 (blue) are shown. Solvent molecules and hydrogen atoms are omitted for clarity. **b** and **d** Relative positions of the major Th1/Y1 unit localized in a region of the fullerene cage. Metal-cage, metal-centroid, and metal-metal distances (in Å) are indicated. **e** Isosurface (±0.002 au) of the spin density distribution for ThY@$D_{3h}$(5)-C$_{78}$. **f** ESR X-band spectra measured for a CS$_2$ solution of ThY@$D_{3h}$(5)-C$_{78}$ under different temperatures (in K) with simulated results represented with smooth solid lines. Calculated (Calc.) EPR spectrum at the DFT level is also plotted, which is downfield 45 G shifted.

ThDy@$D_{3h}$(5)-C$_{78}$, and 3.939 Å for ThDy@$I_h$(7)-C$_{80}$, as well as the metal-cage distances are very close to the experimental values (Table 1). Different from ThDy@C$_{2n}$, a spin-doublet electronic state is found for ThY@$D_2$(10611)-C$_{72}$ and ThY@$D_{3h}$(5)-C$_{78}$. Th and Y ions are located within the $D_2$(10611)-C$_{72}$ and $D_{3h}$(5)-C$_{78}$ cages in the same positions as

the ThDy analogs. Th-Y and metal-cage distances are also shown in Table 1. Th-cage distances are nearly identical for the different sized cages, however, the Th-Y distances within the $D_2$(10611)-C$_{72}$ and $D_{3h}$(5)-C$_{78}$ are about 0.02 Å shorter than for ThDy, therefore there are longer Y-C$_{cage}$ contacts than for Dy-C$_{cage}$.

**Table 1 | Computed electronic and structural data and observed first reduction potentials for ThX@$D_2$(10611)-C$_{72}$ and ThX@$D_{3h}$(5)-C$_{78}$ (X = Dy and Y), ThDy@$C_s$(17490)-C$_{76}$, ThDy@$I_h$(7)-C$_{80}$, and Th$_2$@$I_h$(7)-C$_{80}$**

| | $D_2$(10611)-C$_{72}$ | | $C_s$(17490)-C$_{76}$ | $D_{3h}$(5)-C$_{78}$ | | $I_h$(7)-C$_{80}$ | |
| --- | --- | --- | --- | --- | --- | --- | --- |
| | ThDy | ThY | ThDy | ThDy | ThY | ThDy | Th$_2$[34] |
| Spin (Th) [a] | 0.55 | 0.52 | 0.53 | 0.48 | 0.43 | 0.48 | / |
| Spin (X) [a] | 5.52 | 0.53 | 5.54 | 5.44 | 0.53 | 5.44 | / |
| d(ThX) [b] | 4.187 | 4.169 | 4.209 | 4.151 | 4.137 | 3.939 | 3.817 |
| | (4.166(3)) | (4.183(1)) | (4.193(2)) | (4.135(9)) | (4.144(8)) | (3.932(7)) | 3.816 |
| d(Th-C$_{cage}$) [b] | 2.529 | 2.528 | 2.515 | 2.509 | 2.506 | 2.502–2.528 | 2.547 |
| d(X-C$_{cage}$) [b] | 2.409 | 2.428 | 2.401 | 2.419 | 2.434 | 2.418–2.427 | / |
| $\varepsilon_o$(LUMO) [c] | −2.94 | −3.11 | −3.06 | −3.55 | −3.68 | −3.92 | / |
| $E^{0/-}$ [d] | −1.49 | −1.52 | / | −1.03 | −0.97 | −0.66 | / |
| $\rho_{bcp}$ [e] | 0.108 | 0.109 | 0.105 | 0.106 | 0.107 | 0.132 | 0.234 |
| $\nabla^2\rho_{bcp}$ [e] | −0.178 | −2.03 | −0.175 | −0.165 | −0.194 | −0.186 | −0.510 |

[a]Atomic Mulliken spin densities for Th, Y, and Dy.
[b]Metal-metal and metal-carbon cage distances (in Å).
[c]Energies of the sigma LUMO (beta) orbital (in eV).
[d]First reduction potentials (in V).
[e]Electron density and Laplacian of the electron density at the bond critical points are given in [eÅ$^{-3}$] and [eÅ$^{-5}$], respectively.

The spin density distribution for ThDy@C$_{2n}$ and their individual Mulliken spin populations at metal atoms (-5.5 for Dy and -0.5 for Th), computed for the septet state, show that the unpaired electrons are mainly localized on Dy with a contribution around half an electron at Th (Supplementary Figs. 10 and 13). These data indicate that the Dy and Th atoms have formal oxidation states of 2.5+ and 3.5+, respectively. The ThDy cluster formally transfers six electrons to the fullerene, with a resulting electronic structure of (ThDy)$^{6+}$@(C$_{2n}$)$^{6-}$. Figure 3a shows a rather simplified molecular orbital (MO) diagram for the ground spin-septet state of ThDy@$D_2$(10611)-C$_{72}$, ThDy@$D_{3h}$(5)-C$_{78}$ and ThDy@$I_h$(7)-C$_{80}$ with one electron in the delocalized $\sigma$ orbital. For simplicity, the corresponding MO diagram for ThDy@$C_s$(17490)-C$_{76}$ is shown in the Supplementary Fig. 14a. Supplementary Figs. 15–18 provide the MOs associated with the Dy f$^9$ for the ThDy@C$_{2n}$ series, which are localized on Dy and are found to be lower in energy compared to the $\sigma$-bonding orbital. For ThY@C$_{2n}$, there is also a transfer of six electrons from ThY to the C$_{2n}$ cage. The corresponding MO diagrams for the ground spin-doublet state are also plotted in Fig. 3a, with one electron in the delocalized $\sigma$ orbital. An interesting correlation between the ThX distances and the energies of the $\sigma$ orbital for the different C$_{2n}$ cages was found, with the shortest metal-metal distance showing the $\sigma$ bonding orbital with the lowest energy (C$_{80}$ for ThDy and C$_{78}$ for ThY, Fig. 3). This trend is confirmed by measuring the redox potentials, which are very different from those of EMFs containing pairs of lanthanides as La$_2$ or Ce$_2$ because of different electron occupations of the metal-metal (spd) sigma molecular orbitals, providing more evidence of the Ln-An single electron bond inside fullerene cages (see Supplementary Information).

The computed first oxidation and reduction potentials for ThY@C$_{2n}$ (2n = 72 and 78) show good agreement with the experimental data (see Supplementary Table 10). As expected from Fig. 3, the first oxidation process for both ThY@C$_{72}$ and ThY@C$_{78}$ involve MOs of the fullerene, with the triplet spin-state configuration being, by far, the lowest in energy. The first reduction process is predicted to take place at the Th-Y sigma orbital, which is the HOMO in the singlet spin configuration of the reduced system of ThY@C$_{78}$ (see Supplementary Figs. 22 and 23 and Supplementary Table 11). We predict that the first reduction for all ThX@C$_{2n}$ (2n = 72, 78, and 80; X = Y and Dy) reported in this article occurs at the sigma Th-Ln orbital in line with the correlation between the energies of these orbitals and the first reduction potentials (Table 1).

Similar bonding has been previously reported for some Ln$_2$@C$_{2n}$ species[14]. As shown in Fig. 3 and Supplementary Fig. 11, the $a_1$ orbital exhibits similar electron distributions of s, p and d populations for the two metal atoms, so the single electron is shared by the two metals. Furthermore, Bader's Quantum Theory of Atoms in Molecules[37] was used to characterize the An(Th)-Ln(Dy) interaction. The bond critical point (BCP) postulated by Bader between two atoms is a necessary and sufficient condition for the atoms to be bonded. At the BCP, the corresponding values of electron density $\rho_{bcp}$ and Laplacian of the electron density $\nabla^2\rho_{bcp}$ for ThX@C$_{2n}$ series (X = Dy and Y) are also displayed in Table 1, confirming the presence of an accumulation of charge density in the center of the metal-metal bonds caused by a significant overlap between hybrid spd orbitals of the two metals. Note that we get smaller values than for Th$_2$@$I_h$(7)-C$_{80}$[34], which also shows a smaller Th-Th distance and two electrons in the sigma bond.

## ESR analysis of ThY@$D_{3h}$(5)-C$_{78}$

The ESR spectra of ThY@$D_{3h}$(5)-C$_{78}$ as a function of temperature show a doublet, as expected for an unpaired electron mainly interacting with Y (nuclear spin $I$ = 1/2 of $^{89}$Y, 100% natural abundance), while the nuclear spin of $^{232}$Th is 0 in 100% natural abundance. The smooth solid lines are simulated results using a hyperfine constant $A_{iso}$ = 200 MHz and $g_{iso}$ = 1.825 (for 220, 190 and 150 K) with EasySpin software package based on Hamiltonian (1).

$$\hat{H} = \mu_B B g_{iso}\hat{S} + \hat{S}A_{iso}\hat{I} \tag{1}$$

DFT calculations are in full agreement with experiments, resulting in a calculated $A$ of 206 MHz and a $g$ value of 1.803 (Fig. 2f and Supplementary Fig. 20). The large hyperfine coupling constant indicates that there is a significant amount of spin density located on yttrium[38,39]. The isotropic signal with $g_{iso}$ = 1.825 is somewhat smaller than the $g$ value of a 6d single electron in traditional Th-based monometallic complexes[40]. Below 150 K, no signal was observed due to the frozen motion of ThY@$D_{3h}$(5)-C$_{78}$, which may broaden the spectral peaks. These results indicate that the unpaired electron is not on the fullerene cage but it is mainly confined to yttrium and partially delocalized on thorium, an indication that a single electron bond is formed between Y and Th, consistent with the electronic structure observed in the computations (Supplementary Fig. 13). The hyperfine coupling constant of ThY@$D_{3h}$(5)-C$_{78}$ is very similar to that found for Y$_2$@C$_{79}$N, in which a two-center single-electron sigma bond is also present[39].

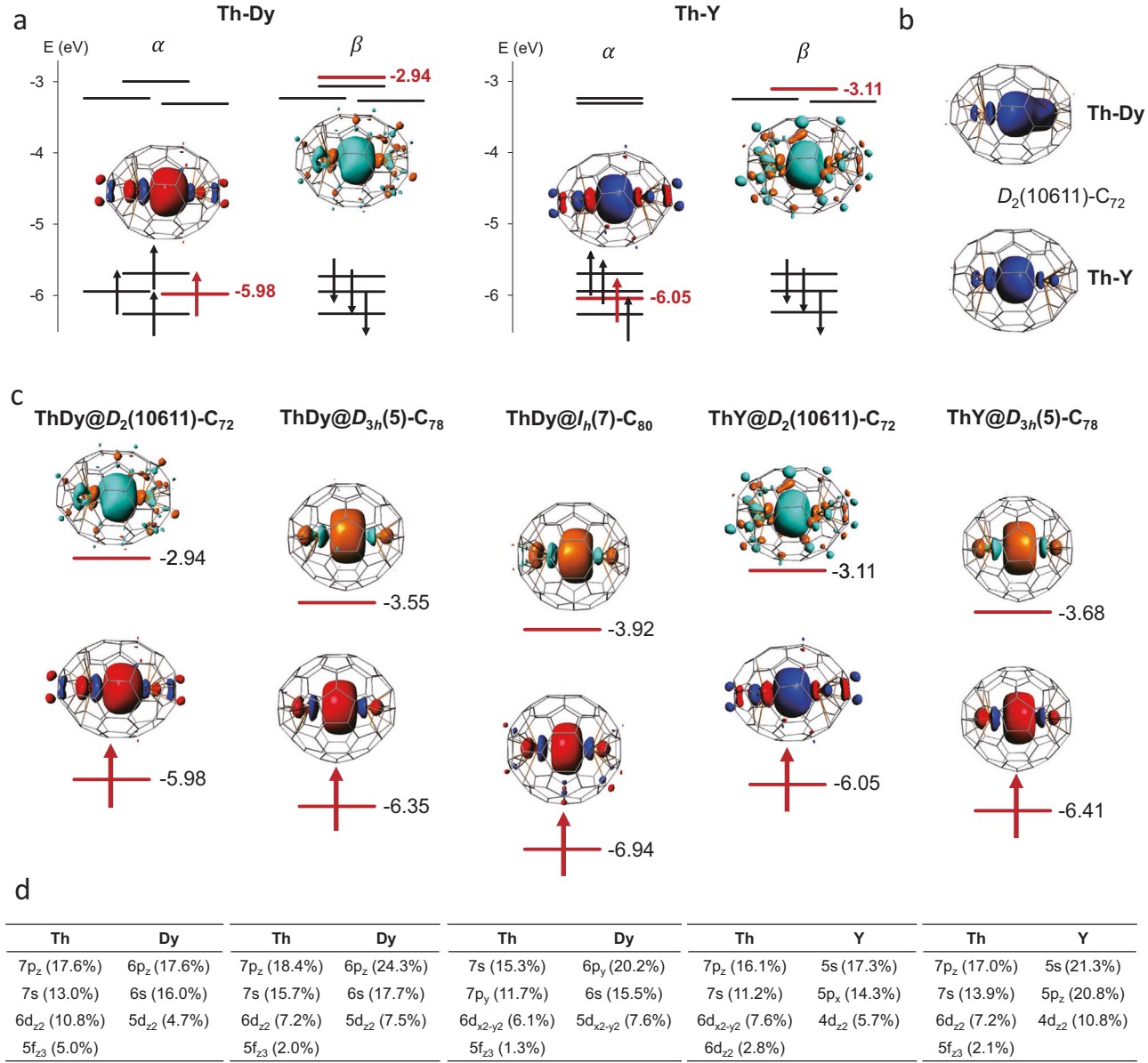

**Fig. 3 | Electronic structures for selected ThX@C$_{2n}$ (X = Dy and Y) systems.**
**a** Molecular orbital (MO) diagrams for ThDy@$D_2$(10611)-C$_{72}$ and ThY@$D_2$(10611)-C$_{72}$. The energy levels of the delocalized sigma orbital $a_1$ (for alpha- and beta-spins) are drawn in red and the associated MO isosurfaces (±0.03 a.u.) with the corresponding MO energy (in eV) are shown on the side. The energy levels for the highest six electrons (in black) from the cage are also represented. For more detailed MO diagrams, see Supplementary Fig. 12. **b** Spin density distribution with an isosurface of ±0.002 au for ThDy@$D_2$(10611)-C$_{72}$ and ThY@$D_2$(10611)-C$_{72}$. **c** MO isosurfaces (±0.03 a.u.) and MO energies (in eV) for the delocalized sigma orbital $a_1$ for alpha- and beta-spins of ThDy@$D_2$(10611)-C$_{72}$, ThDy@$D_{3h}$(5)-C$_{78}$, ThDy@$I_h$(7)-C$_{80}$, ThY@$D_2$(10611)-C$_{72}$ and ThY@$D_{3h}$(5)-C$_{78}$. **d** Molecular orbital contributions of the $\sigma$-type bonding orbital formed essentially by ns, np and (n-1)d metal orbitals.

## SQUID magnetometry of ThDy@$D_{3h}$(5)-C$_{78}$

To further explore the electronic structure of the obtained dimetallo-EMFs, direct current (dc) magnetic measurements were performed for polycrystalline sample of ThDy@$D_{3h}$(5)-C$_{78}$. The temperature-dependent susceptibility data were collected under 1 kOe dc field in the temperature range of 2–300 K. The magnetic susceptibility χ at first remains almost constant as the temperature decreases (Supplementary Fig. 27), while the χT products slowly decrease as temperature changes from 300 K to around 10 K (Fig. 4c). The effect of crystal-field splitting is dominant with further cooling, and χT drops rapidly from 10 K to 2 K due to the depopulation of Stark sublevels. This variation tendency has been reported in many literatures[4,41–43]. Magnetization data were collected at 2, 3, 5, 8, and 10 K in the field range of 0~50 kOe (Fig. 4b). The non-superposition of magnetization curves at different temperatures suggest magnetic anisotropy caused by the crystal field.

Magnetic measurements showed that no hysteresis could be observed even at 2 K with the sweep rate of 500 Oe s$^{-1}$. This is different from what previously observed for Ln$_2$@C$_{79}$N and Ln$_2$@C$_{80}$(CH$_2$Ph) (Ln = Dy, Tb)[19,36], which exhibit exceptionally high blocking temperatures at ca. 20 K. Compared with Dy(III) and Tb(III) ions, the f orbital of Th is more diffuse, and Th is more susceptible to the external crystal field (fullerene cage). The orbital angular momentum of Th is easily quenched, and thus the anisotropy is usually weak, leading to different electronic structure and magnetic behavior from Ln(III) ions. Therefore, the introduction of Th may cause ThDy@$D_{3h}$(5)-C$_{78}$ to exhibit completely different hysteresis behaviors from dilanthanide metallofullerenes. Besides, the Dy-to-ring-centroid distances in ThDy@$D_{3h}$(5)-C$_{78}$ (1.867 Å) is shorter than that in Dy$_2$@C$_{79}$N[36] (average: 1.916 Å) and Dy$_2$@C$_{80}$(CH$_2$Ph)[18] (average: 1.988 Å) (Supplementary Table 13), indicating that the fullerene cage may have a stronger effect on the inner

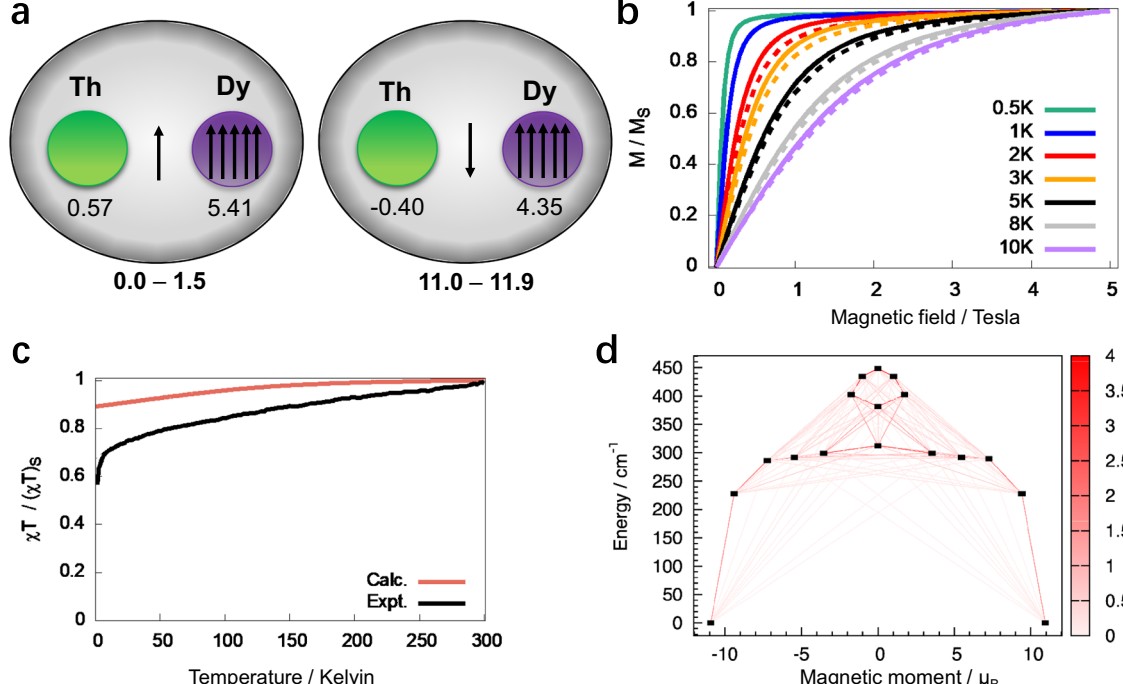

**Fig. 4 | Magnetic properties of ThDy@$D_{3h}$(5)-C$_{78}$. a** Spin distributions and relative energies (in kcal·mol$^{-1}$) for the lowest two spin states determined from CASSCF calculations. Atomic Mulliken spin densities are given for the lowest-energy spin state of each spin configuration. The arrow in the middle of the two metal atoms represents that the unpaired electron in the sigma bond is delocalized between the two centers. **b** Calculated magnetization curves measured at various temperatures (0.5, 1, 2, 3, 5, 8, and 10 K). Corresponding experimental curves at $T$ = 2, 3, 5, 8, and 10 K are plotted in dashed lines. **c** Experimental *vs.* calculated normalized $\chi T$. **d** Low-energy spectrum with transition probabilities visualized as lines of different thicknesses (thicker lines correspond to higher probabilities), the x axis is the projection of magnetic moment upon the main anisotropy axis. Calculations in Fig. 4b–d include spin-orbit coupling.

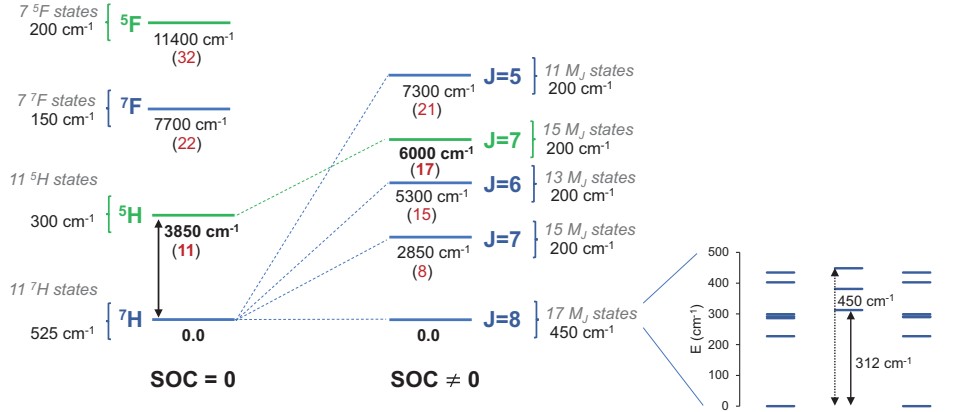

**Fig. 5 | Schematic representation of the lowest-energy states of ThDy@C$_{78}$ with and without including spin-orbit coupling (SOC) at CASSCF level.** The spin-orbit-free (left, SOC = 0) and the spin-orbit (right, SOC ≠ 0) states are represented in blue for the septet spin state and green for the quintet spin state. The number of spin-orbit-free states of each type, described as $^{2S+1}$L as if they were spherical atoms, are indicated. Relative energies in parentheses are in kcal·mol$^{-1}$. The total angular momentum J (spherical atom approximation) is indicated in the SOC ≠0 section. We make the assignment of these atomic-like J values according to the manifold of $M_J$ states for each level. The energy difference between the septet and quintet states (SOC = 0) is in bold, as well as the energy difference between the lowest-energy spin-orbit states derived from the septet and the quintet. For the lowest-energy level (J = 8), the representation of its 17 $M_J$ states with their corresponding energies is plotted on the side showing the energy barrier (312 cm$^{-1}$) and the energy splitting among these 17 states (450 cm$^{-1}$). A scheme of the magnetic moment of Dy$^{3+}$ ($J_{Dy}$ = 15/2) and the spin in the sigma orbital (s = j = 1/2) alignment in the ground state is also shown. For sake of interpretation, we consider that the delocalized electron in the sigma $a_1$ orbital has no first-order angular momentum (l = 0).

Dy ions, which likely reduce the uniaxial anisotropy and cause a much lower blocking temperature.

CASSCF calculations for the ThDy@$D_{3h}$(5)-C$_{78}$ molecule were performed to corroborate the DFT results, as well as to determine the magnetic properties arising from the magnetic anisotropy of the Dy center. We used an active space with eight orbitals and ten electrons and explored the lowest quintet and septet states (Supplementary Table 7). For ThDy@$D_{3h}$(5)-C$_{78}$, in the spin-orbit free description, the spin-septet state is the lowest-energy state, confirming the results obtained from the DFT calculations, and it is followed by the spin-quintet state at 3850 cm$^{-1}$ (11.0 kcal·mol$^{-1}$) higher in energy (Fig. 5 and Supplementary Table 8). The spin distributions of the active orbitals

for the lowest-energy spin states of ThDy@$D_{3h}$(5)-$C_{78}$ are displayed in Fig. 4a and Supplementary Table 8, being the spins parallel for all of the septet states. For the quintet states, the spin of the delocalized electron in the $a_1$ orbital is antiparallel to the five f electrons localized on Dy. At PBE0 level, the energy difference between the high and low spin states is reduced to 1400 cm$^{-1}$ (4.0 kcal mol$^{-1}$), which is rather similar to the value reported for a related digadolinium endofullerene (1220 cm$^{-1}$, 3.5 kcal·mol$^{-1}$)[19]. When the spin-orbit coupling is considered at CASSCF level, the energy difference increases by 2100 cm$^{-1}$ (6 kcal·mol$^{-1}$, Fig. 5). In Fig. 5, we have correlated the spin-orbit states (SOC ≠ 0) with the spin-orbit free states (SOC = 0) computed at CASSCF level. We have named the states as if they were spherical atoms, $^{2S+1}$L, with SOC within a LS coupling scheme, which helps us in the interpretation of the results. Once the SOC is included, the level with J = 8 from septet state is the ground state multiplet (17 $M_J$ states split in 450 cm$^{-1}$ range), followed by J = 7 level from septet state (15 $M_J$ states) at 2850 cm$^{-1}$ (8 kcal·mol$^{-1}$), and the J = 7 level (15 $M_J$ states) from quintet state is found at 6000 cm$^{-1}$ (17 kcal·mol$^{-1}$). The lowest SO-levels show contributions dominated mainly from the SO-free septet states with a slight degree of mixing with quintet states.

Figure 4b compares the calculated normalized magnetization *vs.* the experimental normalized magnetization at 2, 3, 5, 8, and 10 K, resulting in a good agreement between calculations and experiments with a saturation of the magnetization at a magnetic field between 1 and 2 T. In addition, calculated normalized χT curve was obtained and compared with the experimental normalized χT curve (Fig. 4c). Some discrepancies between calculation and experiment were observed in the χT vs T curves at low temperatures. These deviations are challenging to avoid due to the limited sample yield and the difficulty in accurately calculating the magnetic energy levels of 4f or 5f ions[18,43,44]. Nevertheless, from that, we corroborate (i) the spin-septet ground state of this system; and (ii) the high energy of the spin-quintet state that makes it to be non populated at room temperature. Such an energy difference likely comes from the nearly atomic-like Hund coupling between the sigma electron and the unpaired 4f electrons on Dy. As a consequence, the behavior of χT vs T plot is like a high-spin state with magnetic anisotropy in contrast to the χT curves reported for Dy$_2$@$C_{80}$(CH$_2$Ph)[18]. For the latter, two centers (Dy) with magnetic anisotropy are coupled through a ferromagnetic interaction with the spin of the electron in the sigma Dy-Dy bond. Different from our systems, the reported studies of endohedral dilanthanofullerenes[18,23,45,46] all contain two centers with magnetic anisotropy that could explain the larger values computed for the barrier of magnetization (426 cm$^{-1}$ and 582 cm$^{-1}$ for Dy$_2$@$C_{80}$(CH$_2$Ph) and Dy$_2$@$C_{79}$N, respectively, *vs.* 312 cm$^{-1}$ for ThDy@$D_{3h}$(5)-$C_{78}$, see Fig. 4d and Fig. 5. Magnetic studies are underway for other members of the An-Ln@$C_{2n}$ series to evaluate the relevance of the number of magnetic centers and metal-metal distances. Preliminary calculations show that the barrier of 400 cm$^{-1}$ can be easily overcome.

## Discussion

In summary, we report the synthesis and characterization of a series of mixed-valence dimetallo-EMFs, ThDy@$C_{2n}$ (2n = 72, 76, 78, and 80) and ThY@$C_{2n}$ (2n = 72 and 78), featuring single-electron actinide-lanthanide metal-metal bonds. These mixed-dimetallic EMFs were successfully isolated and structurally characterized by single-crystal X-ray diffraction and various spectroscopic methods. Crystallographic characterization unambiguously confirmed that Th and Ln metals (Y, Dy) can be encapsulated inside variable fullerene carbon cages with different size and symmetry and revealed the cage-dependent metal-metal distances. The ESR study of ThY@$D_{3h}$(5)-$C_{78}$ shows a doublet as expected for an unpaired electron interacting with the Y, and a SQUID magnetometric study of ThDy@$D_{3h}$(5)-$C_{78}$ revealed the high-spin ground state of the whole molecule. DFT and CASSCF calculations identify the presence of a single-electron bonding interaction between

Y or Dy and Th, due to a significant overlap between hybrid spd orbitals of the two metals. Moreover, computational studies further suggest that the characteristic flexibility of Th in terms of oxidation states permits its sharing of an electron with the other metal, resulting in a formal oxidation state of +3.5.

The dimetallo-EMFs we report provide the experimental proof of direct lanthanide-actinide metal-metal bonding, which will add to our understanding of metal-metal bonding involving f-block elements. In fact, the six dimetallo-EMFs reported in this article represent only a small fraction of the many possible members of this new An-Ln@$C_{2n}$ family. Our preliminary studies show that single-electron metal-metal bonding can be formed between an array of lanthanides and U or Th. Computational studies predict that the coupling of a single electron in the sigma-bonding orbital with 4f$^n$ and 5f$^n$ on different lanthanides and actinides may give rise to exceptional magnetic properties to the bulk materials, which may help in the design and synthesis of next-generation molecular magnets.

## Methods

### Synthesis and isolation of ThDy@$C_{2n}$ (2n = 72, 76, 78, and 80) and ThY@$C_{2n}$ (2n = 72 and 78)

The carbon soot containing thorium-dysprosium/yttrium-based EMFs was synthesized by the direct-current arc-discharge method. Graphite, mixed with ThO$_2$ and Dy$_2$O$_3$/Y$_2$O$_3$ (molar ratio of Th:Dy/Y:C = 1:1:24) were evaporated in a 200 Torr He atmosphere with the current of 90 A. The resulting soot was extracted with CS$_2$ for 12 h. The separation and purification of ThDy@$C_{2n}$ (2n = 72, 76, 78, and 80) and ThY@$C_{2n}$ (2n = 72 and 78) were achieved by a multistage HPLC procedure (Supplementary Figs. 2–8). Multiple HPLC columns, including Buckyprep-M (25 × 250 mm, Cosmosil, Nacalai Tesque Inc.), Buckyprep (10 × 250 mm, Cosmosil, Nacalai Tesque, Japan), 5PBB (10 × 250 mm, Cosmosil, Nacalai Tesque, Japan), and Buckyprep-M (10 × 250 mm, Cosmosil, Nacalai Tesque, Japan), were utilized in this procedure. Toluene was used as the mobile phase and the UV detector was adjusted to 310 nm for fullerene detection. The HPLC traces and single peaks of MALDI-TOF spectra for purified samples are shown in Supplementary Figs. 2–8.

### Spectroscopic and electrochemical studies

The positive-ion mode matrix-assisted laser desorption/ionization time-of-flight (Bruker, Germany) was employed for mass characterization. UV-vis-NIR absorption spectra were measured in carbon disulfide at room temperature with a Cary 5000 UV-vis-NIR spectrophotometer (Agilent, U.S.). Cyclic voltammetry was performed in 1,2-dicholorbenzene (o-DCB) with 0.05 M (n-Bu)$_4$NPF$_6$ using a CHI-660E instrument.

### X-ray crystallographic study

Black co-crystals of ThDy@$C_{2n}$ (2n = 72, 76, 78, and 80)·[Ni$^{II}$(OEP)] and ThY@$C_{2n}$ (2n = 72 and 78)·[Ni$^{II}$(OEP)] were obtained by allowing the benzene solution of [Ni$^{II}$(OEP)] and the CS$_2$ solution of each sample to slowly diffuse at 4 °C for 2-3 weeks. X-ray diffraction data were collected at 120/130 K using a diffractometer (Bruker, D8 Venture) equipped with a CCD detector. The structures were solved by a direct method and refined with SHELXL-2015[47]. The cif files of the six crystals in this work are shown in Supplementary Data 1 and ORTEP-style illustration with probability ellipsoids are shown in Supplementary Fig. 26.

### ESR and SQUID magnetometry

Continuous-wave (CW) EPR experiments were performed on a Bruker ElexSys E580 spectrometer at the X-band (ω = 9.36 GHz) with the samples dissolved in CS$_2$. The low-temperature environment was achieved by using an Oxford Instruments ESR900 and CF935 liquid helium cryostat. The EPR spectra were all simulated using the

"EasySpin" toolbox based on MATLAB[48]. DC magnetic properties were determined using a Quantum Design MPMS3 VSM magnetometer. The sample was prepared by drop-casting from $CS_2$ solution onto a slice of Al foil (3.224 mg), which is paramagnetic to minimize the background of the sample holder. Fast evaporation of the carbon disulfide afforded a black powder. After that, the Al foil was folded into a small cube and stuck on the inner wall of a plastic straw with very small amount of N grease (less than 1 mg).

## Computational details

Kohn-Sham density functional theory (DFT) calculations were performed with the Amsterdam Density Functional (ADF, v. 2019) package[49] using the PBE0 exchange correlation functional[50–52], in conjunction with all-electron triple-ζ polarized (TZP) Slater-type orbital (STO) basis sets quality[53,54]. Scalar relativistic (SR) zero-order regular approximation (ZORA) was included for relativistic effects[55]. 'D3' dispersion corrections by Grimme were also performed[56,57]. ESR calculations were performed using an all-electron basis set with ADF and simulated with EasySpin package. Single points CASSCF calculations were carried out for the ThDy@$C_{78}$ system using a PBE0 geometry with OpenMolcas[14]. The active space contains eight orbitals and ten electrons. Extended relativistic ANO-RCC-type basis sets were employed, in particular we used the ANO-RCC-VDZ for Dy and Th, and the ANO-RCC-MB for C. Scalar relativistic effects were considered with the Douglas-Kroll-Hess Hamiltonian[37]. Calculations of the magnetic properties for ThDy@$C_{78}$ were performed at the CASSCF/SO-RASSI level of theory with the use of the SINGLE_ANISO module[58].

## Data availability

The X-ray crystallographic coordinates for structures reported in this study have been deposited at the Cambridge Crystallographic Data Centre (CCDC), under deposition numbers CCDC-2108663 (ThDy@$D_2$(10611)-$C_{72}$), CCDC-2108664 (ThDy@$C_s$(17490)-$C_{76}$), CCDC-2108688 (ThDy@$D_{3h}$(5)-$C_{78}$), CCDC-2108689 (ThDy@$I_h$(7)-$C_{80}$), CCDC-2108690 (ThY@$D_2$(10611)-$C_{72}$) and CCDC-2108694 (ThY@$D_{3h}$(5)-$C_{78}$). Copies of the data can be obtained free of charge via https://www.ccdc.cam.ac.uk/structures/. The Source Data files contain spectroscopy, chromatography, DC magnetic data, and xyz data of simulated structures. All other data supporting the findings of this study are available from the corresponding authors on request. Source data are provided with this paper.

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

## Acknowledgements

N. C. thanks the National Science Foundation China (NSFC NO. 52172051, 91961109) and the NSF of Jiangsu Province (BK20200041), the Priority Academic Program Development of Jiangsu Higher Education Institutions (PAPD). A.R.-F., J.M.P., and C.d.G. thank the Spanish Ministry of Science (grants PID2020-112762GB-I00 and PID2020-113187GB-I00), the Generalitat de Catalunya (grant 2021SGR00110) and the URV for support. Y.R.Y thanks the National Natural Science Foundation of China (22301288), the Anhui Provincial Natural Science Foundation (2308085MB31), and the Fundamental Research Funds for the Central Universities (WK2060000051). L.A. thanks the AGAUR for a Beatriu de Pinós fellowship. L.E. thanks the NSF for the generous support of this work under Grant CHE-1801317. The Robert A. Welch Foundation is also gratefully acknowledged for an Endowed Chair to L.E. (Grant AH-0033).

## Author contributions

N.C. conceived and designed the experimental work. Y.J.Y. synthesized and isolated all the compounds. Y.R.Y., Y.J.Y., and Q.Y.M. performed the crystallographic analysis. L.A., Y.R., A.R.-F., C.d.G., and J.M.P. performed the computations and theoretical analyses. S.G., B.W.W., R.S., and Y.H.F. performed the SQUID and ESR magnetometry. Y.J.Y. performed the electrochemical test. N.C., A.R.-F., J.M.P., L.E., Y. J.Y., L.A., S.R., Y.R.Y., Y.S., and M.H.J. co-wrote the manuscript.

## Competing interests

The authors declare no competing interests.
