## [Peer Review File · Nature Communications]

Actinide-Lanthanide Single Electron Metal-Metal Bond Formed in Mixed-Valence Di-metallofullerenesReviewers' Comments:

Reviewer #1:

Remarks to the Author:

It is a pity that the authors were unable to conduct some further measurements due to the limited quantity of the sample. But they did a nice job addressing most of the points that were raised. I'm happy to be able to recommend this for publication in Nature Communications.

Reviewer #2:

Remarks to the Author:

The manuscript by Yan et al. is a revised version of the rejected previous Nature Chem paper. The authors have made some effort to accommodate the substantial criticisms of the reviewers, but mostly put forward reasons not to do the work. In principle the novelty would be sufficient for Nat. Commun. The one criticism that has not been addressed at all is the discrepancy between measured and calculated χT . First of all, the statement of the authors in the response letter "We would also like to emphasize that Fig. 4c does include spin-orbit coupling (SOC $\neq 0$). We have clearly specified it in caption of Figure 4 of the revised version of the manuscript." appears to be false, as the caption of Fig. 4c in full is "(c) Experimental vs. calculated normalized χT ." The text further states that "In addition, similar calculated and experimental normalized χT curves are obtained (Fig. 4c).", where similarity is of course in the eye of the beholder. The explanation for the discrepancy (response to comment 2 of referee 3) is mostly given in terms of experimental challenges. Where is the discussion of the theoretical results? If the authors state that "the sharp decrease in the low temperature range is owing to the intrinsic anisotropy of Dy ions caused by the crystal field" then clearly the theoretical description is wrong, because it is not reproduced by theory. In any case in the same response, I do not understand what the difference is between "Stark sublevels" and "crystal field" in terms of the underlying physical interactions?

So once more I have to recommend rejection.

Reviewer #3:

Remarks to the Author:

In the current manuscript the authors have taken into account my previous comments, so I can accept it as it is.

Response to reviewer's comments:

Reviewer #2:

The manuscript by Yan et al. is a revised version of the rejected previous Nature Chem paper. The authors have made some effort to accommodate the substantial criticisms of the reviewers, but mostly put forward reasons not to do the work. In principle the novelty would be sufficient for Nat. Commun.

Response: We would like to express our gratitude to the reviewer for taking the time to review our manuscript. In the following section, we provide a detailed point-by-point response to the reviewer's comments.

(1) The one criticism that has not been addressed at all is the discrepancy between measured and calculated XT. First of all, the statement of the authors in the response letter "We would also like to emphasize that Fig.4c does include spin-orbit coupling (SOC \neq 0). We have clearly specified it in caption of Figure 4 of the revised version of the manuscript." appears to be false, as the caption of Fig. 4c in full is "c) Experimental vs. calculated normalized XT."

Response: We appreciate the reviewer's reminder and have made the necessary modification to the caption of Fig. 4. Specifically, we have revised the corresponding text 'These calculations include spin-orbit coupling' to read as follows: "Calculations in Figures 4b, 4c, and 4d include spin-orbit coupling."

(2) The text further states that "In addition, similar calculated and experimental normalized XT curves are obtained (Fig.4c).", where similarity is of course in the eye of the beholder. The explanation for the discrepancy (response to comment 2 of referee 3) is mostly given in terms of experimental challenges. Where is the discussion of the theoretical results? If the authors state that "the sharp decrease in the low temperature range is owing to the intrinsic anisotropy of Dy ions caused by the crystal field" then

clearly the theoretical description is wrong, because it is not reproduced by theory. In any case in the same response, I do not understand what the difference is between "Stark sublevels" and "crystal field" in terms of the underlying physical interactions? So once more I have to recommend rejection.

Response:

1. For the discrepancy between measured and calculated χ_T :

a) We acknowledge the reviewer's point that the perception of similarity can vary among individuals. While we consider that there are similarities between the measured and calculated χ_T , we also recognized that deviations in χ_T measurements are an objective matter. Despite our best efforts, we cannot completely avoid them, particularly for metallofullerenes, where yield limitations introduce additional uncertainty. In fact, it is worth noting that this deviation is not uncommon and can also be observed in other types of samples with sufficient yields. (e. g. N. F. Chilton and D. P. Mills, *et.al. Nature*, **2017**, *548*, 439). For such small sample quantity of metallofullerenes, the measured results would be more easily affected by the background (e.g. sample holder). Thus, the reported magnetic susceptibilities or magnetization of metallofullerenes have been all scaled based on theoretical or saturated values (e.g. S.-Y. Xie, A. A. Popov, T. Greber and S. -F. Yang, *et.al. J. Am. Chem. Soc.* **2016**, *138*, 14764; A. A. Popov, *et.al. Nat. Commun.*, **2017**, *8*, 16098), as was done in our work. The correction in this work is based on the measured χ_T at 300 K, and this single-point correction itself may cause bias, so it is hard to guarantee the complete reproduction of scaled results.

b) Theoretical discussion of the discrepancy: Owing to the much smaller energy scale of magnetic fine energy levels than that of bonding, it is more challenging to calculate the magnetic energy levels precisely, especially excited states. Especially the behavior at low temperatures is very sensitive to small changes in relative energies of the lowest-energy spin-orbit states. This is why we chose multi-configuration based method rather than DFT to calculate the magnetic properties. In theory, this bias of χ_T values is

pervasive and difficult to overcome due to the complexity of the electronic structure of 4f and 5f ions. For 4f ions, the calculated χT values are relatively more reliable, but there still be some biases, especially in the low temperature region (e.g. N. F. Chilton and D. P. Mills, *et. al. Nature*, **2017**, 548, 439; L. Norel and J. R. Long, *et. al. Angew. Chem. Int. Ed.* **2018**, 57, 1933; A. A. Popov, *et. al. Chem. Sci.*, **2017**, 8, 6451-6465; M.-L. Tong and R. A. Layfield, *et. al. Chem. Eur. J.* **2023**, 29, e202300567; Lai Feng, Peng Jin, Chunru Wang and Taishan Wang, *et. al. Nanoscale*, **2019**, 11, 186112 -18618). The introduction of 5f ions in our molecule make the question more complicated. The calculated χT vs T plot is rather sensitive to the calculated energies of the 17 M_J states derived from the spin-orbit coupling (Fig. 4d). The fine energy levels of low-lying excited states are very difficult to be computed precisely with the current *ab initio* calculations for large molecules, especially including metal ions. To provide further clarification, we have included a brief discussion which states the discrepancies and acknowledges the challenges associated with both experimental measurements and theoretical calculations (Page 14).

2. For the difference between "Stark sublevels" and "crystal field":

The splitting of energy levels caused by an electric field is known as Stark effect. Here, the fine energy levels generated by crystal-field splitting are called Stark sublevels. We apologize for any confusion caused by our previous response letter. We agree that the sentence "*The continuous decrease of magnetic susceptibility with temperature decreasing in the high temperature range is owing to the thermal depopulation of the Stark sublevels, while the sharp decrease in the low temperature range is owing to the intrinsic anisotropy of Dy ions caused by the crystal field*" was not expressed appropriately and could mislead readers regarding the concepts of "Stark sublevels" and "crystal field." We appreciate the reviewer's critical reminder and have made the following correction:

In the temperature range of 2-300 K, magnetic susceptibility χ at first remains almost constant as the temperature decreases (Response Fig. 1), while the χT products slowly decrease as temperature changes (Fig. 4c). The effect of crystal-field splitting is

dominant with further cooling, and χT drops rapidly due to the depopulation of Stark sublevels. This variation tendency has been reported in many literatures (e. g. N. F. Chilton and Jeffery R. Long, *et. al. J. Am. Chem. Soc.*, **2023**, *145*, 1572; N. F. Chilton, B. G. Harvey and Jeffery R. Long, *et. al. Science*, **2022**, *375*, 198; Y. -Z. Zheng, *et. al. Chem. Commun.*, **2019**, *55*, 9355; N. F. Chilton and D. P. Mills, *et. al. Nature*, **2017**, *548*, 429; B. -W. Wang and S. Gao, *et. al. Dalton Trans.*, **2016**, *45*, 8149; J. L. Tian, J. K. Tang and S. P. Yan, *et. al. Chem. Eur. J.*, **2012**, *18*, 2484-248). And we have added Response Fig. 1 as new Supplementary Fig. 27. The corresponding changes have been made in the main text (Page 12).

Response Fig. 1. Experimental temperature dependence of χ for ThDy@D_{3h}(5)-C₇₈

Reviewers' Comments:

Reviewer #2:

Remarks to the Author:

The second revised version has now sufficiently addressed previous concerns and I recommend acceptance.